# Transcutaneous Functional Electrical Stimulation Controlled by a System of Sensors for the Lower Limbs: A Systematic Review

**DOI:** 10.3390/s22249812

**Published:** 2022-12-14

**Authors:** Layal Chaikho, Elizabeth Clark, Maxime Raison

**Affiliations:** 1Lab of Intelligent Biomechanics, Robotics, and Rehab Technology (LIBRTy), Department of Mechanical Engineering, Polytechnique Montréal, P.O. Box 6079 Station Centre-Ville, Montréal, QC H3C 3A7, Canada; 2Institute of Biomedical Engineering, Polytechnique Montreal, P.O. Box 6079 Station Centre-Ville, Montréal, QC H3C 3A7, Canada; 3Independent Researcher, Montreal, QC H3C 3A7, Canada

**Keywords:** FES, sensors, rehabilitation, nonlinear control, systematic review

## Abstract

In the field of transcutaneous functional electrical stimulation (FES), open-loop and closed-loop control strategies have been developed to restore functions of the lower limbs: walking, standing up, maintaining posture, and cycling. These strategies require sensors that provide feedback information on muscle activity or biomechanics of movement. Since muscle response induced by transcutaneous FES is nonlinear, time-varying, and dependent on muscle fatigue evolution, the choice of sensor type and control strategy becomes critical. The main objective of this review is to provide state-of-the-art, emerging, current, and previous solutions in terms of control strategies. Focus is given on transcutaneous FES systems for the lower limbs. Using Compendex and Inspec databases, a total of 135 review and conference articles were included in this review. Recent studies mainly use inertial sensors, although the use of electromyograms for lower limbs has become more frequent. Currently, several researchers are opting for nonlinear controllers to overcome the nonlinear and time-varying effects of FES. More development is needed in the field of systems using inertial sensors for nonlinear control. Further studies are needed to validate nonlinear control systems in patients with neuromuscular disorders.

## 1. Introduction

Functional electrical stimulation (FES) has been an active field of research for more than 60 years [1]. It has traditionally been used to improve functional motor abilities in patients presenting neurological conditions such as spinal cord injury (SCI) [2], stroke [3], cerebral palsy [4], spina bifida [5], multiple sclerosis [6] or muscular dystrophy [3]. In these conditions, the patients present either complete or partial paralysis or muscle weakness.

FES is a modality used to either assist or totally activate the contractions of weak and/or paralyzed muscles. It is also a modality used as a temporary support in a rehabilitation phase or as a treatment modality integrated into an orthosis, namely a hybrid orthosis. Thus, the use of an FES in the field of rehabilitation is aimed at both assistive and restorative effects. The FES control strategies are divided into two hierarchical levels presented below (Figure 1):

(1)High-level control (HLC): This one includes open-loop control and finite-state control (FSC). Open-loop control does not necessarily require the use of sensors. However, if it does, it is an ON-OFF control, which is a high-level control. It often requires continuous inputs from the user, who must then devote her/his attention to the operation of the FES device [7]. This method is currently widely used in patients with paresis due to its ease of implementation and use by users [8]. However, technically, this type presents with output errors, lack of sensitivity to external disturbances (e.g., spasticity), internal uncertainties (e.g., muscle fatigue), and overstimulation of muscle fibers [9]. These can affect the precision and duration of the executed movement [7]. FSC requires the use of sensors. This type of FES system performs a predefined stimulation sequence in an open loop when a condition of interest measured by sensors is satisfied. It is mainly used in patients with foot drop [10].(2)Low-level control (LLC): This one refers to closed-loop control. This type is a promising approach for trajectory tracking control [11]. LLC can be linear (LLLC) or nonlinear (NLLLC). Factors that influence FES-induced limb movements include:Progressive state of muscle fatigue [12] in the face of synchronous [13] and non-physiological [11] recruitment of motor units.Temporal change in the physiological properties of the motor units, namely excitability, contractility, and conductivity [14].Additional voluntary input of the patient [15].Level of spasticity [16].Electromechanical delay (EMD) [17].

Therefore, the dynamic model describing lower limb movements is nonlinear, time-varying, and uncertain [18]. Several research studies on the last five have thus been led to focus on the development of a closed-loop control strategy [17,19,20,21,22,23,24,25,26,27,28,29,30,31,32]. The manipulated variables in closed-loop FES systems are the stimulation wave parameters (e.g., current amplitude or pulse width). The controlled variables are the neuromuscular (e.g., muscle force), the kinematic (e.g., joint angle) or kinetic (e.g., joint torque) variables.

In the FES field of use, a modeling method can help to establish a relationship between the FES inputs and the outputs, measured by sensors. This relationship can then be used in the development of a control strategy. The choice of sensor depends on the controlled variable and influences the overall development of FES systems. The control strategy, which requires sensors, involves these two hierarchical levels: low-level control and high-level control. Figure 2 illustrates an LLC system in the context of FES. This highlights the relationship, showing the link between these three concepts: 1. the FES model, 2. the sensor feedback, and 3. the control strategy.

Several reviews addressing control strategies used in FES systems have been published to date [8,10,11,13,33,34]. To our knowledge, there has been no literature review summarizing the following three concepts simultaneously: 1. modeling of FES-induced response, 2. type of sensor, and 3. control strategy. There are no findings of a literature review grouping both FES induced complex movements sequences and simple movements. Complex movements sequences are involved in activities such as walking, standing up, maintaining posture, and cycling. Examples of simple movements are knee or ankle motions in the sagittal plan without ground impact.

The main objective of this systematic review is to provide a state-of-the-art survey of emerging and current solutions in term of control strategies using sensors. It focuses on studies carried out on transcutaneous FES systems for the lower limbs in an assistive and restorative context in the field of rehabilitation. The sub-objectives (SO) are grouped as follows:

SO1: List and compare the different methods used to model the FES-induced muscle response or biomechanics response.

SO2: List and compare the different types of sensors used in FES systems.

SO3: List and compare different control strategies for FES systems.

SO4: Establish the relationships between SO1, SO2 and SO3.

SO5: List the validation method of the FES system: simulation, healthy subjects, and patients.

## 2. Materials and Methods

This systematic review follows the Preferred Reporting Items for Systematic Reviews and Meta-Analyses (PRISMA) guidelines [35].

### 2.1. Summary of the Development Phases of FES Systems

Different stages of development are required for the design of FES systems for lower limb movements, as follows:(1)The development of signal processing technique for the feedback information extracted from the sensors, used for high- and low-level control, if any.(2)The development of an algorithm for the detection of the intention of movement or phase detection, for high-level control, when applicable.(3)The presence of a method for calibrating the FES parameters, if any.(4)The development of a dynamic model or an experimental identification model of FES.(5)The development of a control strategy, including the development of appropriate low-level controllers.(6)Numerical simulations, experiments on healthy subjects, and/or experiments on subjects with motor deficits.

### 2.2. Data Sources

The strategy to select the articles is carried out in six steps:

Step 1. CONCEPTUAL PLAN: The analyzed documents were screened from two databases: Compendex and Inspec. The concept plan contains terms related to functional electrical stimulation as well as terms related to control aspects (with keywords “Control” that consider open-loop, FSC and closed-loop control). The following combination of words was used: (((((“electrical stimulation”) AND (“Closed loop” OR Control* OR “Closed-loop”)) WN LA))

Step 2. ELIMINATION BY LANGUAGE: Only articles written in English were considered.

Step 3. ELIMINATION OF DUPLICATES: Duplicates that were caused by one of the following two factors were rejected: (1) multiple versions of the same article found in the same database, or (2) the same article found in both databases.

Step 4. ELIMINATION BY VOCABULARY: The following controlled vocabularies were excluded from the search: neurosurgery, sensory feedback, agricultural robots, microelectrodes, surgical implants, iron, rats, mammals, controlled drug delivery, surgery, animals, amines, cardiology.

Step 5. ELIMINATION OF DOCUMENTS THAT ARE NOT ARTICLES: Documents that are neither journal nor conference articles (e.g., theses, books, or literature reviews) were rejected.

Step 6. ELIMINATION BY CRITERIA: The titles and abstracts of the articles from step 5 (or their full text, if necessary) were screened one-by-one by a reviewer (L.C.) to identify eligible articles. Those ones that do not meet the following criteria were rejected:Involved limbs: lower limbsType of FES electrodes: transcutaneous electrodesInvolved living beings: healthy human subjects and/or human patientsDevelopment and/or presentation of a control strategyUse of sensors or generation of simulation results.

Further, a study that used sensors and developed a control strategy without prior modeling/identification was also considered for this review.

Articles were not considered if they reported only data related to clinical outcomes. The initial data base screening was performed on 8 December 2021, and an update was carried out on 13 May 2022.

### 2.3. Data Extraction

The extracted variables were: year of publication, type of sensor, type of movement, stimulated muscles, modeling and/or identification method, choice of controlled variable, method of calculating controlled variable, control strategy, and validation method. When the experiment was performed on patients, the neurological or neuromuscular condition was also indicated. Some papers included sensors that are not directly attached to body segments. This was the case of encoders in the motors of a hybrid orthosis, sensors attached to pedal cranks, and sensors attached to mechanical assistance devices. To simplify data analysis and reduce analysis biases, these papers were grouped separately in some cases.

Additionally, in this systematic review:Only the main sensors used for extracting feedback signals were presented. Sensors used in the phase of the system identification were therefore not considered.The type of controlled variable was the last variable to appear at the output of the control system. In other words, the controlled variables in the inner loops of the control system were not mentioned in the main results of the overview.Only the control strategy used for FES was presented. Thus, for motorized hybrid systems, the control strategy used for motors was not presented.

### 2.4. Data Analysis

Due to the heterogeneity in the selected articles about the types of chosen sensors, the protocol used, and the targeted movements, the results of this review were not pooled into a meta-analysis. A descriptive synthesis of the results was carried out. The mean value and standard deviation (SD) were applied on some data. The χ^2^ statistical test was used to compare proportions (with α = 0.01).

The processed variables were the following:Target joint: If the target joint is not explicitly stated in an article, it was simply deduced by the action of the stimulated muscles.Directionality of the movements induced by the FES: determined according to the stimulation of antagonist–agonist muscles acting on the same joint.Determination of the hierarchic level of control, i.e., high and/or low level: In this review, high level control includes open loop control (via inverse model-based control and ON-OFF control), and FSC. As for low level control, it included any control that allows trajectory tracking control or the reaching of a target value (which can be linear or nonlinear).Determination of the type of low-level control: In this review, a nonlinear controller was judged “nonlinear” only if it was used to compensate the system model’s nonlinearities, or if it is explicitly indicated in the article.

The modeling/identification methods were classified as follows: the dynamic models, including the muscle activation, muscle contraction and/or multibody models, vs. experimental identification models, including transfer function and non-parametric identification. The sensors were gathered and classified as follows: neuromuscular sensors, kinematic sensors, and kinetic sensors. The control strategies were gathered and classified as follows: linear low-level control (LLLC), nonlinear low-level control (NLLLC), high-level control (HLC), high-level control and linear low-level control (HLC+LLLC), and high-level control and nonlinear low-level control (HLC+NLLLC). The combination of HLC and LLC is used, for example, when a detection of phase (using HLC) and a trajectory tracking (using LLC) are both required.

## 3. Results

Figure 3 shows the results from the article selection strategy, leading to the selection of 135 articles. Figure 4 presents the overall results for each category of sensors for electromyography (EMG) (Figure 4a), inertial sensor (Figure 4b) and kinetic sensor (Figure 4c). Amongst the neuromuscular sensors, only EMGs are represented as the main ones. Amongst the kinematics sensors, only inertial sensors are represented as the main ones (see Table 1 for the most recent sensors in FES systems).

### 3.1. Modeling Methods of FES Systems

Figure 5 shows the type of sensors used in the studies according to the type of models.

#### 3.1.1. Dynamic Models

In many studies, Hammerstein’s model was used to represent the dynamics of muscle activation and those of muscle contraction [19,36,37,38,39,66]. A state model with Kalman’s filter is usually used to identify this model [36,37,38]. Let us note that without using Hammerstein’s model, a dynamic model composed of a 1st-order activation model based on Ca2+ release, a 1st-order muscle contraction model, and a 2nd-order multibody dynamic model was developed [67]. This model established a relationship between the stimulation intensity and the angular position of the knee. More recently, a dynamic model composed of a 3rd-order nonlinear activation model with saturation, a muscle contraction model considering the effects of fatigue, and a 2nd-order nonlinear planar two-segment (i.e., thigh and leg) model was proposed [68]. This model was multiple-input multiple-output, which established a relationship between pulse widths (PW) and angular positions of the hips and knees.

Some studies have used Hill’s muscle model for the dynamics of muscle contraction [29,69,70,71]. Without using a specific Hill model, neural networks (NN) were used to approximate the nonlinear parameters of a 1st-order muscle contraction model, establishing a relationship between knee muscle torques to PW [72].

In 1990, Durfee and Dilorenzo (1990) proposed a dynamical model based on a simple inverted pendulum, i.e., one segment, to model the leg movement stimulated by FES [73]. Most studies that developed a multibody dynamics model using equations of motion (e.g., Newton–Euler equations) applied it to the simple flexion-extension movement of the knee [17,18,31,40,41,67,72,73,74,75,76,77,78,79,80,81,82,83,84] or ankle [85]. In walking, a multibody dynamic model of ankle motion was developed [29]. For more than one joint in walking, a multi-body multibody humanoid model [42] and a biarticular multibody model [68] were proposed. Since postural balance requires the activation of multiple joints, some studies developed a dynamic multibody model [32,86,87,88,89,90,91]. In the standing position, the human body was modeled as a single [92,93,94,95], double [86,87] or triple [89,90,91] inverted pendulum. Movement equations for cycling were also proposed based on the cycle-rider dynamic model [96,97,98,99,100,101,102,103,104,105]. Furthermore, a real-time deep neural network (DNN) was used to estimate the nonlinear and uncertain dynamics of each leg during motorized cycling [106]. Complex biomechanical models using state models were proposed in some studies for posture [86,90] and for simple knee extension [69,71]. Simple three-segment biomechanical models for movement from sitting to standing were highlighted in the development of controllers based on mechanical energy conservation [107,108]. In other studies, a dynamic model of a humanoid was developed using the 4D Nastran^®^ dynamic simulation software for cycling [109,110] and for walking [42,111].

#### 3.1.2. Experimental Identification Models

Previously, Stanic and Trnkoczy (1974) developed a linear mathematical model of the ankle-antagonist muscle pair [112]. This was possible by the identification of a 1st- order transfer function between FES voltage and joint torque by applying a proportional-integral (PI) controller. Jaeger (1986) proposed a 2nd-order transfer function between PW and ankle muscle torques [89]. Similarly, in 2000, Ferrarin and Pedotti proposed a dynamic model driven by a 2nd-order nonlinear differential equation to find a relationship between stimulation intensity and knee joint torque [74]. Several studies in this review used this approach [41,78,113]. One study included muscle torque and delay caused by spasticity [78]. Recently, an approach based on the Takagi–Sugeno fuzzy method was proposed to determine a one-pole transfer function relating knee muscle torque to pulse width [113]. Pseudo-random binary sequence (PRBS) signals were used to identify local linear functions at each operating point between pulse width and ankle joint torque [43,93]. Considering delays, a 1st-order transfer function with the introduction of an EMD made it possible to establish a relationship between stimulation intensity and muscle torque generated by FES [87]. A relationship between muscle activation and muscle torque induced by FES was described by a critically damped 2nd-order transfer function with delay [114]. A transfer function between current FES amplitude and joint angles, namely the knee and ankle, was proposed using an iterative optimization algorithm [44].

Without using a dynamic model or transfer function, three studies meeting the inclusion criteria have used a nonparametric identification method. A nonlinear autoregressive network with exogenous inputs (NARX) model was proposed to determine the angular position of the knee joint under FES [115,116]. Furthermore, on-line fuzzy system identification was proposed to approximate the model nonlinear functions [45].

### 3.2. Type of Sensors Used in FES Systems

The neuromuscular sensors found include electromyography (EMG; Section 3.2.1) and electroencephalography (EEG; Section 3.2.2). The kinematic sensors include inertial sensors (Section 3.2.3), and conventional angular sensors, such as potentiometers, goniometers, laser sensors, and encoders (Section 3.2.4). The kinetic sensors include force sensors and torque sensors (Section 3.2.5). The extracted information was processed to form a controlled variable. Table 1 shows the average year of publication for each type of sensor. The three most used sensors recently are: 1. kinematic sensors for cycling, 2. inertial sensors, and 3. EMG. Figure 6 shows the type of sensors in the studies according to the type of movement (a) to the system complexity (b) and to the directionality induced by FES (c).

#### 3.2.1. Electromyography

The analysis of EMG signals provides information on muscular activation. When EMGs are integrated into FES systems, they enable detection of the movement intention by extracting the volitional EMG signal (vEMG). This has been made possible by using the envelope of the vEMG signal [46], the residual signal between two M-waves which are muscle electrical signals induced by FES [47], and an adaptive filter applied on the mixed EMG signal [48]. The vEMG signal was used to determine the optimal time to trigger electrical stimulation [21] and to calculate the stimulation intensity in real-time by using the average threshold crossing method [49].

When used in a closed-loop control system, the use of M-waves belonging to the FES-evoked EMG signal (eEMG) would modulate the FES-induced muscle response. The combined use of eEMG, torque muscle modulation [15,37,50], and muscle activation modulation was proposed from its mean average value [36].

When the paralysis is incomplete, voluntary contractions also contribute to the movement in addition to the FES-induced contractions. Thus, a parallel filter-based algorithm was implemented to separate the mixed EMG signal into the eEMG signal and the vEMG signal [15]. The evoked and voluntary joint torques were then estimated using a radial-based NN algorithm.

#### 3.2.2. Electroencephalograms

A recent development in FES systems provided integration with EEG that converts brain signals into commands to FES [117]. In these systems, EEG could measure neurophysiological activity in the motor cerebral cortex to detect movement intentions using event-related desynchronization signals [117]. The application of a common spatial pattern filter to the data to extract features and the use of linear discriminant analysis for classification were used to detect the intention to extend the legs [51] and to move the lower limbs [20].

#### 3.2.3. Inertial Sensors

As wearable sensors, inertial measurement units (IMU) can usually be integrated with accelerometers, gyroscopes, and magnetometers. They measure linear accelerations, angular velocities, and the earth’s magnetic field vector in a three-dimensional orthogonal local coordinate system. They allow estimation of body segment orientation by combining multiple pieces of information using optimal sensor fusion algorithms. Due to the large measurement errors associated with the perturbations in the magnetic field, the magnetometer signal was not used in any of the investigated studies.

The combination between different inertial sensors, rigidly connected to articulated body segments, and the kinematic constraints allow for the calculation of joint angles. Inertial sensors were used in recent studies on systems controlling the angular position of a joint through FES [14,23,41,44,45,52,53,54,55,56,118]. Inertial sensors were also used in FES systems for cycling to control crank speed [21]. With inertial sensor-based systems, finite state algorithms were proposed to detect the phases of a movement, such as the phases of the gait cycle [28,52,57] combined to force sensitive resistors (FSR) [44,55,56,58], or the phases in the movement of standing up [59].

#### 3.2.4. Conventional Kinematic Sensors

Traditionally, angle sensors are used to provide feedback on angular positions, joint angular velocities, and angular accelerations of joints. Sensors connected to human bodies that were proposed in the studies of this review include goniometers [23,41,60,66,80,107,108,115,119,120,121,122,123,124,125,126,127,128,129], potentiometers [22,77,95,112,130,131,132,133], inclinometers [134], high-torque drive timing belt [7], and laser sensors [91]. As some studies have proposed to integrate FES into robotics, various kinematic sensors attached to mechanical assistive devices were proposed. For example:The sensors embedded in a leg extension machine were optical encoders [17,18,75,76,79,83].The sensors embedded in an inverted pendulum standing apparatus were laser displacement sensors [92,94].The sensors embedded in a seesaw standing up motion machines were goniometers [135].The sensors embedded in systems such as LOKOMAT^®^ (HOCOMA, Switzerland) walking robots were potentiometers [136].The sensors embedded in Wobbler’s posture maintenance apparatus were potentiometers [43].A 3D motion capture system was also used [134,137].

The sensors attached to the hybrid systems in this review were the encoders embedded in the motors of the motorized orthoses [25,32,61,90,114,136,138,139,140]. The type of encoders used (i.e., mechanical, optical, electromagnetic, or inductive) was not reported for all studies. In cycling, encoders attached to the pedal cranks were used as sensors to detect angular position [96,97,98,99,100,101,102,103,104,105,141,142,143]. Encoders were also used in a rowing exercise [9].

In several studies, simple derivation by differentiation of angular position was applied to obtain joint angular velocity [108,122,123,127] or crank velocity [97,98,99,100,101,102,103,104,105,141,142] as controlled variables. Joint position sensors were used to obtain the 3D position of the center of mass, which then served as a controlled variable [91,144]. When the dynamic parameters of muscle contraction were sufficiently identified, it was possible to extract the joint torques from the joint angles through inverse dynamics [93].

#### 3.2.5. Kinetic Sensors

The type of contraction, i.e., either isometric or dynamic, affects the choice of the type of force sensors to be used. Some studies have proposed closed-loop control of joint torque during isometric contractions [43,51]. However, the sensors used to measure torque during isometric contractions (e.g., load cells) cannot be used to measure torque during dynamic contractions. Therefore, sensors with transducers, such as dynamometers and electrogoniometers, were proposed for the dynamic cases to measure the joint torques [37,38,39,40,60,92,94]. Proposing the use of a proprioception signal to activate FES, a dynamometer placed on a Bowden cable was used [62].

For maintaining posture, plantar force plates were proposed to measure the ground reaction force and control the joint torque for each ankle by FES [87,93].

When detecting phases of gait cycle for high-level control, synchronization is often achieved by using force sensors. FSRs [24,44,58,63,91,119,125,128,140,145], strain gauges [119,128], and simple switches [64,65] were used by a finite-state rule-based control to estimate gait phase to trigger stimulation accordingly.

### 3.3. Control Strategies of FES Systems

The control strategies (LLLC, NLLLC, HLC, HLC+LLLC and HNL+NLLLC) reported in this review differ according to the type of activity, such as walking, standing up, maintaining posture, and cycling. Simple flexion-extension movements of limbs, including thigh, leg, and foot movements, are also reported. Figure 7a shows the control strategies distribution depending on the type of sensor. Figure 7b shows the choice of the controlled variable regarding the type of sensor. Table 2 shows the number of articles with the three most recent sensors between 2012 and 2022 used in a nonlinear control strategy.

#### 3.3.1. Simple Flexion-extension Movements of a Limb

Table 3 summarizes the control strategies for simple flexion-extension movements of a limb in HLC, LLLC and NLLLC. The following paragraphs describe in detail the content and common thread of this table.

According to Table 3, in open-loop control, which is an HLC, a proposed, easy-to-use method for determining stimulation intensity as needed, such as joint torque or angular position, is to apply an identification model [80]. vEMG signal was used to trigger FES on the tibialis anterior muscle for ankle dorsiflexion [48]. EEG signal was used for the detection of the intention of knee extension in FES system [117].

Several studies were reported for closed-loop linear control, which is an LLLC (Table 3). The proportional-integral-derivative (PID) controller was used for simple linear control [23,41,73,78,80,82,112,120,121]. When the system identification was known, a root locus approach was used [41,62]. The focus of some studies was on incorporating a delay-related term into the PID controller to compensate for EMD delays [75,82]. In some studies, a control strategy were formulated to suit for movements that were contaminated by voluntary contractions, such as virtual reference feedback tuning (VRFT) [137,148]. This approach involves a PID controller. It is necessary to account for external delays and disturbances. In some works, control strategies were based on the predictive control (PC) approach, which relies on real-time feedforward correction [19,36,50,71,114].

Several studies were about closed loop nonlinear control, which is an NLLLC (Table 3). To overcome the problems associated with unknown and uncertain model parameters, studies were performed using identification-based control [77], adaptive control based on adjustable PID parameters [80,132,146], reference model adaptive control [66] and adaptive control using Lyapunov–Krasovskii functions [17,40,76]. For accurate trajectory tracking, another study considered adaptive control with a robust integral of the sign of error (RISE) [18]. When an accurate mathematical model of the FES process is not available, a fuzzy logic controller (FLC) was used to account for nonlinearities [22,53,54,81,88,111,113,149]. Due to external and parametric perturbations, some studies considered nonlinear sliding mode control (SMC) [45,68,69,71,78,85,130,132,136]. To eliminate the nonlinearity of the contraction dynamics and the recruitment dynamics, a nonlinear control method with two inverse compensation units was presented [67]. To relieve muscle fatigue, a new asynchronous stimulation technique for the quadriceps with multiple FES channels that deliver electrical current at a lower frequency was recently proposed [83]. The electrical pulses from each channel were controlled by a RISE controller and were out of phase with each other.

#### 3.3.2. Walking

In HLC, an ON-OFF controller based on a finite state algorithm was used for FES-induced gait movements [59,64,65,126,128,129,145]. A vEMG signal was used for the detection of the intention of movement to trigger FES on the tibialis anterior during walking [47]. A finite-state-based algorithm for detecting seven phases of the gait cycle was adopted to trigger stimulation, with considering delays [57].

In HLC+LLLC, PID controller [81,140], a PI controller [28], a PD controller [29], and a P controller [14,56] were used in some studies for trajectory tracking of one or more joints during walking.

Recently, in NLLLC, an SMC [29] and point-to-point repetitive control [26] were used to control the ankle during walking. In another study, FLC for controlling the knee joint during walking was proposed [42].

In HLC+LLLC, an iterative learning control (ILC) algorithm was used to control the angular position of the ankle during walking [52]. Similarly, in HLC+NLLLC, a NN enhanced ILC (NN-ILC) was used to control the angular position of the knee joint during walking [32]. Long- and short-term memory neural networks (NN-LSTM) were applied to predict the EMG signal used to modulate the required stimulation intensity for knee motion in gait movement [24]. In addition, an NN structure was used to determine the stimulation intensity of five muscles during walking as a function of the angular position of the hips, knees, and ankles [125].

#### 3.3.3. Standing Up

A method in LLLC, called ONZOFF control, was proposed to stimulate the quadriceps, gluteal, and hamstrings muscles during the transition from sitting to standing and vice versa [134]. The ONZOFF approach determined stimulation patterns according to a 2^d^-order switching curve defined in a knee angle state space as a function of knee angular velocity.

In, HLC, without a trajectory tracking, an ON-OFF controller embedded in a finite state machine was used [124,129]. In some studies, an ON-OFF switching curve introduced into a state space of knee angle position as a function of knee angle velocity was used [107,108,122,123]. A control structure based on the principle of patient-driven motion reinforcement (PDMR) was developed. In this approach, the motion initiated by the upper body effort is used to calculate the required stimulation intensity for the hip, knee, and ankle muscles [88,124,135]. Furthermore, specifically with EMGs, the envelope of the vEMG signal was used to generate an FES stimulation pattern to the quadriceps and gluteal muscles [46].

In HLC+LLLC, a PID controller was used for trajectory tracking of the knee joint during the transition from sitting to standing [95].

#### 3.3.4. Maintaining Posture

All reported studies with maintaining posture were about LLLC. A simple PID controller was used to control the ankle joint [89,92], to control the knee joint [107,124,133], and to control the 3D center of mass position [90] in the standing position. Furthermore, a PI controller was used to control the knee joint [60] and a PD controller was employed to control the ankle joint [91,94,144] in the standing position. A pole placement-based controller was introduced to control ankle stiffness [93]. In some works, linear-quadratic (LQ) control [86,87] and linear-quadratic-gaussian (LQG) optimal control [39] for ankle motion control in the standing position were adopted.

#### 3.3.5. Cycling and Rolling

In LLLC, to control the pedal crank speed, PID [141], PI [149], and PD [142] controllers were used.

In NLLLC, to control the pedal crank speed, a FLC [21,109,142], a SMC [102], a passivity-based ILC [104], a controller using Lyapunov–Kravoskii functions [97,105], a controller based on Lyapunov analysis [101,103], and a controller based on NN [106] were proposed. To control the joint torque, a controller using passivity-based repetitive spatial learning was considered [100]. To control the pedal crank angle, a RISE adaptive controller was proposed [96].

In HLC, when the cycling phase was detected, an ON-OFF controller for the relevant muscles was proposed [143]. An ON-OFF controller was employed for some phases of rowing by using a switching curve [9].

### 3.4. Methods of Validation of FES Control Strategies

Figure 8 shows the distribution of the last level of validation method for FES control strategies reported from the 135 studies, as presented in Section 3.3. In thirty-seven studies (or 27%), the control strategy was experimented through simulation. In thirty-nine studies (or 29%), the control strategy was experimented on healthy subjects. In fifty-nine studies (or 44%), the trials were carried out with patients. Among these, 42 studies were completed with SCI patients.

## 4. Discussion

This systematic review is a complement to other reviews. The content in the sensor method section complements recent reviews from Schauer et al. (2017) on sensing muscle contraction and motion [11] and from Gil-Castillo et al. (2020) on sensing methods used in FES to correct drop foot [34]. As well, from the control strategy section, complements of information are brought upon recent reviews from Melo et al. (2015) on technical developments of FES to correct drop foot [8] and from Ibitoye et al. (2018) [33] on control strategies in standing. This systematic review brings updated content to the past review from Braz et al. published in 2009 on control FES of standing and walking after SCI [150]. The review from Schauer et al. presented the use of sensors (inertial, EMG, and bioimpedance sensors) and feedback control in FES systems for the upper and lower limbs. However, their section on lower limbs presented information within the limits of the field of gait movement and cycling. An exhaustive review about nonlinear control strategies was not presented at that time. The review from Gil-Castillo et al. described the use of inertial sensors, neuromuscular sensors, and FSR to detect gait phase in FES systems in foot drop. However, not all types of kinetic sensors were covered, the content being presented within the limits of documenting on ankle motion during walking, without focus on control strategies. The focus of the Ibitoye et al. review was on control strategies for standing, so the use of sensors and dynamic models/experimental identification were not reported. The review from Melo et al. was carried out within the limits of looking at strategies for controlling ankle motion in patients with foot drop. The present systematic review of 135 studies provides an overview of previous and recent knowledge in transcutaneous FES models, sensing techniques for controlling FES, and control strategies. It focuses on all lower limb movements. There is a goal of highlighting future research direction for specific applications. However, the present systematic review is carried out within the limits of providing a descriptive list of elements that were reported in studies. The present systematic review does not reach a conclusion about which are the best control strategies and sensors for any given movements/contexts. There is missing information from studies about procedures, and results in robustness and accuracy regarding control strategies. Considering the wide variety of methods used among the studies (including modeling approach, sensor choice, control strategy, and validation method), the authors are not providing a more complete quantitative analysis in the scope of this review. For example, it was difficult to compare the results of a study using FES to stimulate the tibialis anterior during walking (with EMG sensors) with another study that stimulated the quadriceps for a sit-to-stand movement (with inertial sensors). The present review also did not focus on the assessment of effectiveness of the combination of sensors. Additionally, one of the key issues in transcutaneous stimulation is the correct placement of the electrodes, which was not the focus of this review.

Regarding the methodology, a limitation to the present review can come from the little number of databases that were used, i.e., two. Thus, the brought-up profiles of the article (technical vs. clinical studies) might be affected. The analysis was performed regardless of the quality and type of article, such as journal or conference paper. Indeed, no quantitative quality assessment of the 135 studies was performed. Despite these limitations, the authors believe that no high bias in the results should be generated since the main goal of this systematic review was not to analyze the best methods and techniques in FES systems, but to provide a large overview of content about the matter in the domain of rehabilitation. The analysis in this review did not consider the dependence between the studies and the same affiliation of researchers. This limitation can lead to non-negligible biases in the quantitative analysis. An important risk of bias is the heterogeneity of methods among the 135 papers, which did not allow for the quantitative comparative analysis to obtain relevant results on robustness and accuracy depending on the choice of the control strategy and type of sensor used. In addition, the chronological evolution is a limitation considering articles from 1974 to 2022. To lower risk of bias, the authors specifically analyzed the use of the three most recent sensors, and the tendency related to the use of nonlinear controllers was established over the past decade (Table 2).

### 4.1. Specific Objective #1: Modeling Method

In most studies listed, complex dynamic models were used to establish a relationship between the controlled variable and the manipulated variable (Figure 5). In the case of gait movements, these models specifically require consideration of the dynamics of muscle activation, the dynamics of contraction, and the biomechanical model of gait. In that case, anthropometric parameters should be known. Additionally, to mathematically model a system with several unique parameters per individual, future works could focus on the use of a black-box model based on neural networks. The reason being that there are strong nonlinearities and time-varying parameters. Moreover, since muscle fatigue is an inevitable nonlinear process during FES muscle contraction, this characteristic would be needed in the dynamic model as proposed by Ahmed et al. (2018) [151].

### 4.2. Specific Objective #2: Type of Sensor

The choice of the appropriate sensor depends on the type of controlled variable (Figure 4). Neuromuscular sensors are suitable for modulating muscle activity using the eEMG signal. They are suitable for detecting movement intentions, either via electrical motor cortex using EEGs or via electrical muscle activity using EMGs. They are advantageous to avoid the implementation of EMDs in FES controllers. Kinematic sensors are suitable for extracting joint angle measurements of which the trajectory could be specified for low-level control. These ones allow detection of gait cycle phases using a finite-state algorithm. Particularly, inertial sensors are wearable and therefore suitable for complex movements such as walking. Kinetic sensors are suitable for modulating either muscle force or joint torques. These variables also imply a shorter rise time than joint angles when used in control systems. Particularly, FSRs are used to detect the phases of gait cycles.

Lightweight, portable, and real-time measurement systems are emerging as potential technologies for everyday situations, replacing traditional kinematic measurement sensors for FES. As shown in Table 1, two types of sensors are increasingly used for FES control: Inertial sensors and EMGs. The latest traditional kinematic sensors are mainly used for cycling (Table 1).

On the one hand, the methods used for estimating the joint angle with inertial sensors showed a significant improvement over conventional angle sensors, a result in line with Poitras et al. (2019) [152]. However, this method requires the alignment of two coordinate systems: the absolute/inertial coordinate system and the anatomical relative coordinate system. This alignment creates additional complexity for signal processing and can cause calibration errors.

On the other hand, EMGs are increasing and more frequently reported in the literature in cases of integrating eEMG signals into low-level control. Indeed, an internal closed-loop control based on eEMG signals allows to modulate the required stimulation intensity according to the required muscle force. Additionally, the power spectrum of the M-wave could be a good indication of muscle fatigue [19]. Challenges occur from EMG signals being contaminated by the electrical current from the FES stimulator. These artifacts limit the robustness of EMG signal processing. To overcome this, a signal processing technique to remove artifacts is required. For example, a method based on IIR notch filters could be used, as proposed by Rantanen et al. (2018) [153]. Although several studies have exploited the M-wave of the eEMG signal, future works could focus on identifying the differences between eEMG signals from isometric contractions and those from dynamic contractions under FES. Conventional kinematic sensors, which do not require complex and robust signal processing, readily provide kinematic information of lower limb joints, although resolution and accuracy are limited. Recently, Broneira Junior et al. (2021) [51], proposed the use of mechanomyography (MMG) as a sensing technique in FES systems, to obtain feedback of muscle mechanical signals for FES and to provide information on muscle fatigue. However, this study did not involve the development of a control strategy.

### 4.3. Specific Objective #3: Control Strategy

The authors considered evidence that nonlinear controllers (LLLC and HLC+NLLLC) are now used at the expense of traditional linear controllers (Figure 7a). Indeed, linear controllers such as the classical PID controller are questioned. They alone are not suitable for practical FES applications, since they perform poorly in controlling FES-induced muscle contractions when nonlinearity effects are considered.

Among nonlinear control strategies, the commonly used ones in FES systems are adaptive control, RISE control, FLC, PC, and SMC (Table 3). Adaptive control and RISE control are mostly used to overcome problems related to unknown and uncertain model parameters. PC is often used to compensate for delays, such as computational time and EMD. SMC provides fast tuning and good trajectory tracking performance. FLC is known for its ability to control a complex nonlinear system such as FES without having to develop a mathematical model. These types of control can also be enhanced by neural networks.

An important note here is that among the three most recent sensors, the use of inertial sensors as primary sensors for nonlinear control is still increasing (Table 2). Thus, this highlights an existing gap between recent advances in the use of inertial sensors in FES and advances in nonlinear control of FES.

### 4.4. Specific Objective #4: Relationships between SO1, SO2, and SO3

#### 4.4.1. Link between Modeling Method (SO1) and Type of Sensor (SO2)

In dynamic models, the choice of the last hierarchical level (e.g., activation, contraction, multibody models) influences the choice of a controlled variable. Consequently, this choice of a controlled variable directly influences the type of sensor: neuromuscular, kinetic, or kinematic (Figure 4). However, there are exceptions when a relationship between two variables is established in the modeling phase. This relationship requires the use of two different sensors, one of which is not used to calculate the controlled variable in the control strategy. This is the case in [24], where an inertial sensor was used to predict muscle activation by using a neural network and eEMG in the training step. Thus, here, EMG was not used as a sensor in the FES control strategy to extract a controlled variable, but it was only used in the modeling phase. Other examples are in [37,38], where eEMG were proposed to predict muscle torque using dynamometers with Hammerstein’s model. Thus, here, the dynamometers were not used as sensors in the FES control strategy.

#### 4.4.2. Link between Type of Sensor (SO2) and Control Strategy (SO3)

The choice of the controlled variable–i.e., muscle activation, muscle contraction force, muscle torque, joint torque, or joint angle)–could affect the robustness of the controller. Indeed, the robustness is influenced by a multitude of factors, notably by the latency, defined as the delay between the electrical pulse and the measured signal. Indeed, the latency of measuring muscle activation/contraction (e.g., by EMG) is less than the one of measuring joint torque (e.g., by transducers), which in turn is less than the one of measuring joint angle (e.g., inertial sensors) [19]. As shown in Figure 7b, within the high number of studies included in this review, most closed-loop FES control strategies are set between stimulation intensity and joint angle (72 articles) for use in daily lives, measuring and processing joint angle is more practical than joint torque or muscle force.

#### 4.4.3. Link between Modeling Method (SO1) and Control Strategy (SO3)

There is a link between modeling/identification (SO1) and control strategy (SO3). The modeling of the FES-induced response is often required for the development of control strategies. This modeling is used for two purposes: 1. development of identification-based controllers that could consider the nonlinearities, time-varying and uncertainties (e.g., adaptive control), and the stability characteristics (e.g., root-Locus control) 2. optimization of controller parameters through simulations. These purposes justify the importance of an appropriate choice of modeling method in FES systems.

According to Figure 6b, the focuses of most studies were on the stimulation of muscles acting on solely one joint. Considering muscle activities and movements with multiple degrees of freedom is required when it comes to functions such as maintaining posture and standing up. Most of the models in this review are rather based on a simple inverted pendulum. According to Figure 6c, most studies have not stimulated the antagonist muscle pair acting on a joint. This causes a unidirectional movement. Overshooting due to the unidirectionality of the FES-induced movement can occur [107]. Muscle and joint co-working could further be studied.

The details and complexity of FES models could be increasingly reported in the field of control strategy. First, muscle fatigue is an internal source of uncertainty in FES systems. Few studies have considered it in their identification method with direct parameters. Secondly, muscle stiffness and antagonistic muscle activity are other internal source of uncertainties of FES systems that act on time-varying and nonlinear features. Regarding a control strategy, considering the variability of muscle stiffness, which can come from the increase of muscle tone or synergy [16], was carried out in a minority of studies. Muscle fatigue and stiffness are to be considered when developing nonlinear controllers for FES. Thirdly, reflexes, such as spastic reflexes or spinal reflexes, and voluntary contractions are the main external perturbations acting on the FES system. The proportion of studies on the control strategy tested in patients is less than half (Figure 8). Future works could focus on developing control strategies based on patient’s conditions. Finally, EMD is known to cause instability during closed-loop control. In a minority of studies, EMD delays were considered in the dynamic models and/or control strategies. Moreover, EMD was shown to vary with muscle fatigue state [17]. The variability of EMD adds complexity to the current technical challenges associated with FES, which could be the focus of future studies.

### 4.5. Specific Objective #5: Validation Method

It can be observed that fewer studies were carried out with patients for clinical validation, which may reflect the challenges research teams face in having access to patients. According to Figure 8, for most studies (40/57) where systems such as closed-loop FES were tested on patients, it was for cases with SCI. Focus of future works could be on experiences in patients with other disorders. In future development for clinical use, the day-to-day variations of the physiological characteristics of the muscle fibers and the cumulative therapeutic effect in the patient, such as enhanced restorative phase, could be considered.

## 5. Conclusions

The objective of this systematic review was to provide a state-of-the-art survey of emerging and current solutions regarding transcutaneous FES systems models (SO1; Section 3.1), sensing techniques used (SO2; Section 3.2), and control strategies (SO3; Section 3.3). The relationship between these components (SO4) and the clinical transfer (SO5; Section 3.4) were also reported.

The first main result suggested that the use of inertial sensors in FES systems for complex human lower limb movements is increasing (SO2). The main advantage of inertial sensors is their portability for practical use in daily activities.

The second main result suggests that there is a lack of focus from recent studies using inertial sensors on the matter of nonlinear controllers to compensate nonlinear effects of FES-induced muscle response (SO3).

The third main result showed that EMGs are also considered as promising sensors in FES systems (SO2). The main advantage of EMGs is that they provide information about the muscle activation response when accurate feedback to the control system is needed. However, these ones require complex signal processing to adapt to volitional contraction and to suppress artifact from FES current.

The fourth main result showed that researchers are currently developing nonlinear controllers to overcome the nonlinear and time-varying effects of FES (SO4).

The last main result reveals that there is a lack of trials on patients with neuromuscular disorders other than SCI (SO5).

A significant perspective for future development could be towards FES models adapted to patient’s neuromuscular conditions and the testing of a nonlinear control strategy in these patients. Further quantitative studies are needed to provide information on an optimal sensor and choice of a control strategy. Findings from this review may be helpful in identifying new research objectives.

## Figures and Tables

**Figure 1 sensors-22-09812-f001:**
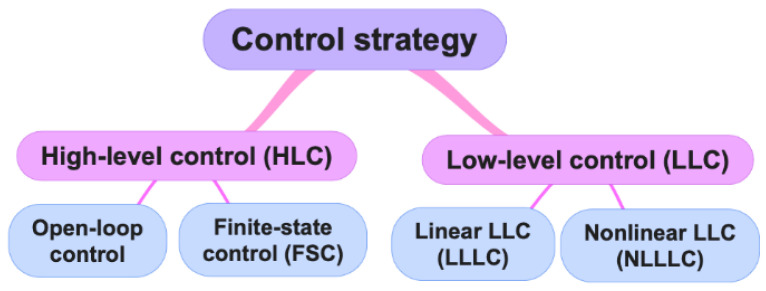
Hierarchical levels of FES control strategies and their subdivisions.

**Figure 2 sensors-22-09812-f002:**
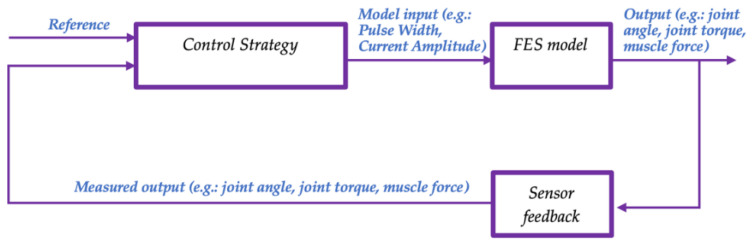
Link between the three concepts in an LLC in the context of FES: (1) FES model (2) Sensor feedback (3) Control strategy.

**Figure 3 sensors-22-09812-f003:**
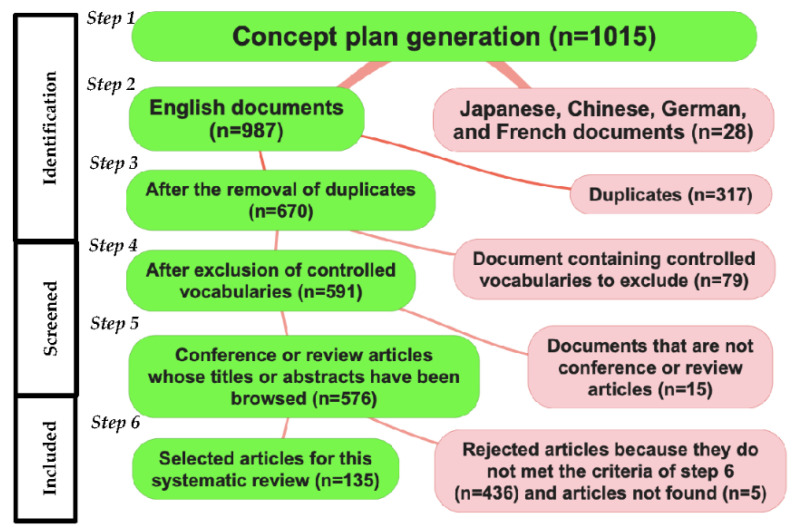
Article selection strategy. The green boxes justify the considered articles. The red boxes justify the ejected articles.

**Figure 4 sensors-22-09812-f004:**
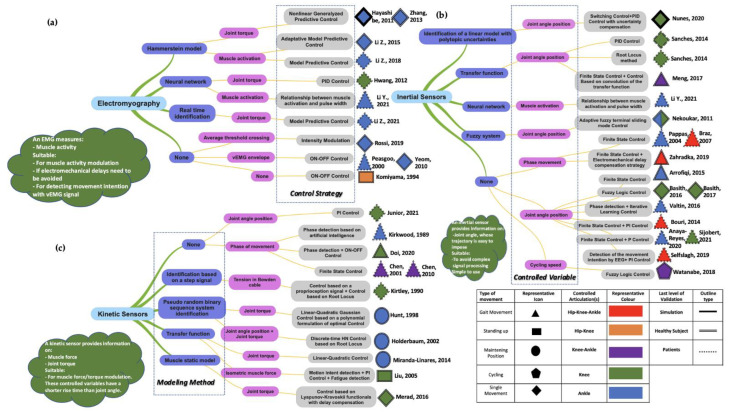
Overall overview for each category of sensor: (**a**) Electromyogramphy; (**b**) Inertial sensors; (**c**) Kinetic sensors. 1st Column: Modeling method; 2nd Column: Controlled variable; 3rd Column: Control Strategy. P: Proportional; I: Integral; D: Derivative [14,15,19,20,21,23,24,36,37,38,39,40,41,42,43,44,45,46,47,48,49,50,51,52,53,54,55,56,57,58,59,60,61,62,63,64,65].

**Figure 5 sensors-22-09812-f005:**
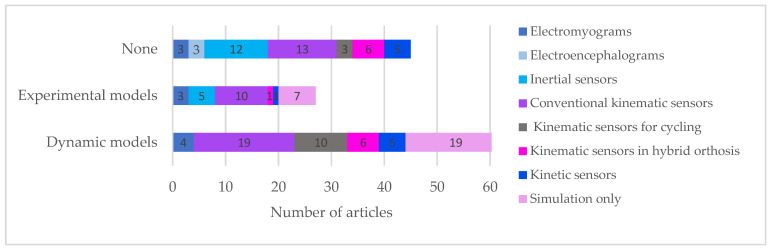
Dynamic vs. experimental models depending on the type of sensor.

**Figure 6 sensors-22-09812-f006:**
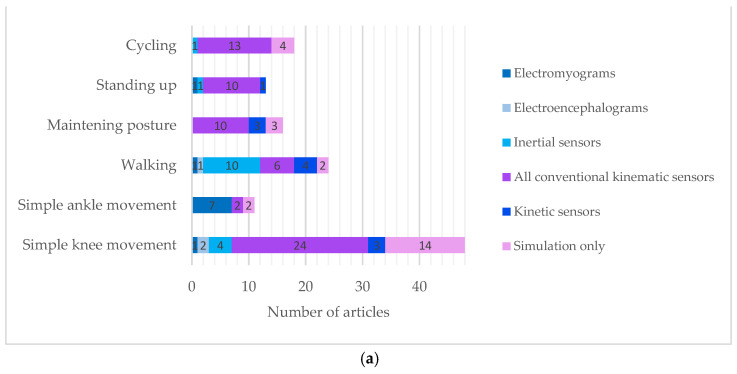
Type of sensors used in the studies according to: (**a**) The type of movement; (**b**) The system complexity; (**c**) The directionality induced by FES.

**Figure 7 sensors-22-09812-f007:**
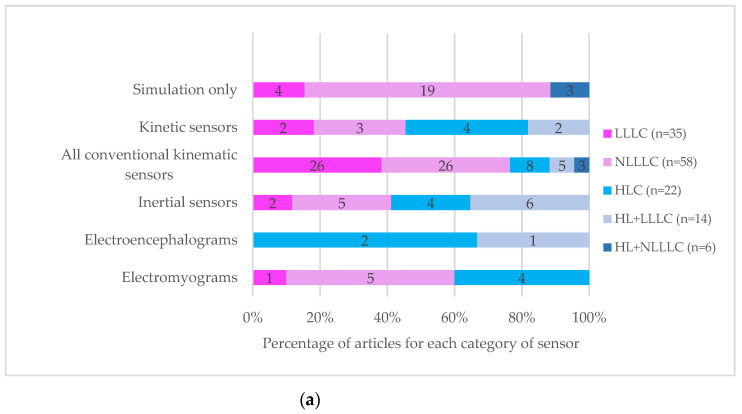
Control strategies description: (**a**) Control strategies distribution depending on the type of sensor; (**b**) Choice of the controlled variable regarding the type of sensor. LLLC: Linear low-level control. NLLLC: Nonlinear low-level control. HLC: High-level control. HLC+LLLC: High-level control and linear low-level control. HLC+NLLLC: High-level control and nonlinear low-level control.

**Figure 8 sensors-22-09812-f008:**
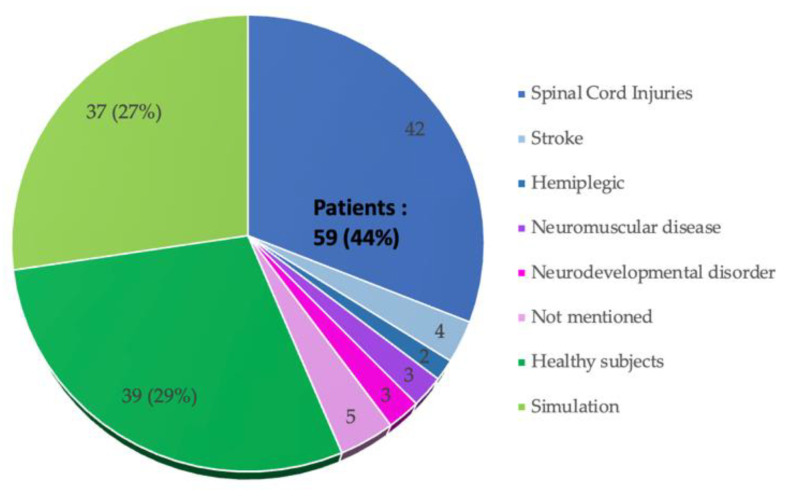
Last level of the experimental method.

**Table 1 sensors-22-09812-t001:** Average year of publication for each type of sensor used in the FES system.

Type of Sensor (n = Number of Articles)	Average Year of Publication (±SD)
Electromyography (n = 10)	2011 (±8) *
Electroencephalography (n = 3)	2016 (±7)
Inertial sensors (n = 17)	2016 (±5) *
Conventional kinematic sensors (n = 42)	2002 (±10)
Kinematic sensors for cycling (n = 13)	2017 (±7) *
Kinematic sensors in hybrid orthosis (n = 13)	2006(±12)
Kinetic sensors (n = 11)	2004 (±11)
Simulation only (n = 26)	2011 (±8)
**All publications (n = 135)**	**2008 (±11)**

* Indicates the three most recent sensors with enough studies (n > 5). It excludes simulation studies.

**Table 2 sensors-22-09812-t002:** Linearity of the control strategy of the three most recent sensors.

Type of Sensor (n = Number of Articles Between 2012 and 2022)	Number of Articles with Nonlinear Controller
Kinematic sensors for cycling (n = 11)	8
Inertial sensor (n = 14)	4 *
Electromyography (n = 6)	4
**Total recent articles (n = 62)**	**37**

* Indicates that the result is significatively different from the overall tendency (using χ^2^ statistical test with α = 0.01).

**Table 3 sensors-22-09812-t003:** Control strategies for simple flexion-extension movements of a limb.

LLLC										
PID	P	PI	PD	I-PD	RLC	PID-DC	VRFT	LQC	PC	FFMC
[23,41,73,78,80,82,112,120,121]	[14,93]	[61,146]	[82,132,147]	[146]	[41,62]	[75,82]	[137,148]	[78,115]	[19,36,50,71,114]	[138]
**NLLLC**										
Identification-based control	AC-PID	RMAC	AC-LKF	RISE	NN-RISE	NN-AC	NN-PID	FLC		NN-FLC
[77]	[80,132,146]	[66]	[17,40,76]	[18]	[70,79,84]	[131]	[116,149]	[22,53,54,81,88,111,113,149]		[149]
Nonlinear PC	SMC	Nonlinear control with two inverse compensation units			Dynamic gradient-based control	Input-output feedback linearization control		Asynchronous stimulation technique-RISE		
[37,38,69]	[45,68,69,71,78,85,130,132,136]	[67]			[31]	[27]		[83]		
**HLC**										
Inverse model of the identification-based control		Detection of the intention of movement by vEMG				Detection of the intention of movement by EEG				
[80]		[48]				[117]				

P: Proportional; D: Derivative; I: Integral; RLC: Root-locus control; PID-DC: PID with delay compensation; VRFT: Virtual reference feedback tuning; LQC: Linear-quadratic control; PC: Predictive control; FFMC: Feed-forward model; AC-PID: Adaptative control with adjustable PID; RMAC: Reference model adaptative control; AC-LKF: Adaptative control based on Lyapunov–Kravoskii functions; RISE: Robust integral of the sign of error. NN-RISE: RISE enhanced with neural network; NN-PID: PID enhanced with neural network; FLC: Fuzzy logic control; NN-FLC: FLC enhanced with neural network; SMC: Sliding mode control; vEMG: Volitional electromyography; EEG: Electroencephalography.

## Data Availability

Not applicable.

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
