# Peer review of "Transcutaneous Functional Electrical Stimulation Controlled by a System of Sensors for the Lower Limbs: A Systematic Review"

_sensors, 2022, doi:10.3390/s22249812_

Round 1

Reviewer 1 Report

The work contains editorial errors:

Line 646  :

Author Response

Thank you for reading!

Best Regards,

Reviewer 2 Report

The paper reviewed transcutaneous functional electrical stimulation (FES), open-loop and closed-loop control strategies have been developed to restore activities of the lower limbs. The paper should be paid more attention in several aspects:

1.     It should provide more details in Figure 2 to show the advantage of overall results for each category of sensor with different control model.

2.     Nonlinear Control for FES should be paid more attention.

3.     Trend in sensors with different functionability should be discussed in the manuscript.

4.     The English requires more attentions throughout the manuscript. The authors should check the English presentation throughout the manuscript. I believe more clear writing could help Authors to express their findings more effectively.

Author Response

Thanks for reading.

Best Regards,

Reviewer 3 Report

The main objective of this systematic review is to provide a state-of-the-art survey of emerging and current solutions in term of control strategies using sensors. It focuses on transcutaneous FES systems for the lower limbs in an assistive and rehabilitative context. FES is of great importance in restoring brain function, and the methods of its development are extremely important. The currently observed development of sensors creates great opportunities for the use of specialized and objective methods of therapy even in home therapy. Therefore, I believe that the work presented for review contributes significantly to the perception of FES and the possibility of standardizing the methods of its application.

Author Response

Thank you for reading

Best Regards,

Reviewer 4 Report

Overall, the review approach a wide topic without selecting one specific area (e.g. application or sensing modality) and without selecting the papers based on any quality indicator. This resulted in a work where a large number of works are included but few quantitative information can be obtained.

1)    The introduction section should be restructured. It is indeed focused on the concept of control strategies, (main aim of the review is to provide a survey of state of the art “control strategies” (line 73)). Are you including both open loop and closed loop control strategies? From the introduction it seems so but the keyword chosen (lines 106 and following) are only focused on closed-loop strategies.

Furthermore, while the main aim in line 73 is on control strategies, just a few lines before (line 67 and following) you stated that you are interested in modeling of FES-induced response (not better explained) together with type of sensors and control strategies. The authors should introduce the concepts before and make explicit the link between the three of them. Why should a further review with all this aspects linked be needed?

Finally, the title refers only to rehabilitative application while in the introduction you refer to “assistive and rehabilitative context”. The reader at this point is confused on what he/she should expect.

2)    The high majority of works with FES are based on a control strategy. Not all of them describe the control strategy (even a closed loop one) by using the words “closed loop”. Nevertheless, the authors chose the keyword “close loop” as criteria to include papers. This is a definition which is not necessarily used when reporting a FES-based study, which in fact have a closed-loop control strategy. Multiple relevant papers are then not included in the current review due to this limitation. As examples for both closed loop and open loop FES rehabilitation: 

-       Ambrosini E, Peri E, Nava C, Longoni L, Monticone M, Pedrocchi A, Ferriero G, Ferrante S. A multimodal training with visual biofeedback in subacute stroke survivors: a randomized controlled trial. Eur J Phys Rehabil Med. 2020 Feb;56(1):24-33. doi: 10.23736/S1973-9087.19.05847-7. Epub 2019 Sep 26. PMID: 31556542.

-       Klauer C, Ferrante S, Ambrosini E, Shiri U, Dähne F, Schmehl I, Pedrocchi A, Schauer T. A patient-controlled functional electrical stimulation system for arm weight relief. Med Eng Phys. 2016 Nov;38(11):1232-1243. doi: 10.1016/j.medengphy.2016.06.006. Epub 2016 Jul 5. PMID: 27397417.

3)    The title itself suggests that the authors want to focus on closed loop strategies only (“controlled by a system of sensors”). This aspect should be clarified and the focus of the review should be better defined

4)    The concept of “top three mostly recent sensor” is unclear to me. How is this ranking defined?

5)    The modeling methods (Specific objective 1) is not really integrated into the review. The link between it and the other aspects (control strategies most importantly). Which is its added value within the review?

Minor review:

6)    Why are you including 135 articles in the review but only 40 papers are included in figure 2. Why?

7)    Line 480. Thirty nine (have) studies…

8)    Line 518 generated results

Author Response

Thank you for reading

Best Regards,

Round 2

Reviewer 2 Report

The manuscript is well revised, and it could be accepted for publication.